# Characterizing urban landscapes using very-high resolution satellite imagery to predict *Ae. albopictus* larval presence probability in public spaces

Claire Teillet[1]*, Héloïse Pottier[1], Rodolphe Devillers[1], Alexandre Defossez[2], Thibault Catry[1], Alexandre Kerr[3], Frederic Jean[4], Gregory L'Ambert[4], Nicolas LeDoeuff[4], Emmanuel Roux[1,5]

1 UMR Espace-Dev, IRD, Université de Montpellier, Université de Guyane, Université de La Réunion, Université de Perpignan, Montpellier, France, 2 UMR TETIS, Université de Montpellier, AgroParisTech, Cirad, CNRS, INRAE, Montpellier, France, 3 Direction Information Géographique, EID Méditerranée, Montpellier, France, 4 Direction Technique, EID Méditerranée, Montpellier, France, 5 International Joint Laboratory Sentinela, IRD, University of Brasília, Oswaldo Cruz Foundation, Brasília, Brazil, Rio de Janeiro, Brazil and Montpellier, France

* claire.teillet@ird.fr

## Abstract

The global spread of *Aedes albopictus* raises growing public health concerns due to its role in transmitting dengue, chikungunya, and Zika. In southern France, the increase in imported dengue cases and local transmission underlines the urgent need for effective vector control. While efforts primarily target private breeding sites, public spaces also contribute notably to larvae presence. Understanding the impact of urban landscapes on the distribution of breeding sites is crucial for optimizing vector control strategies, identifying high-risk areas, and reducing mosquito populations. This study aims to investigate how urban landscapes impact the distribution of *Ae. albopictus* larvae in public spaces, with a focus on storm drains and telecom cable chambers in Montpellier, France. Very high-resolution satellite imagery was used to characterize urban landscapes through textural analyses of spectral indices. Environmental bias was assessed by analyzing the representativity of sampled breeding sites within the diverse urban landscapes. Species distribution models (SDMs) were built, their predictive accuracy was evaluated, and an ensemble model was created to predict larval presence across the study area. SDMs predicted a high probability of larval presence in the western and northeastern parts of Montpellier, with low uncertainty. The most influential variables for predicting larval presence were the mean of Normalized Difference Vegetation Index (NDVI), texture indices from both NDVI, brightness index (BI), and the panchromatic image. Urban vegetation significantly influences larval presence, although higher vegetation index values correlate with a decreased probability of larval occurrence. Additionally, the combination of vegetation and urban structures plays a crucial role in determining the presence of *Ae.*

**Data availability statement:** The data that support the findings of this study are openly available in the DataSuds repository (IRD, France) at https://doi.org/10.23708/TTICOT. Data reuse is granted under a CC-BY license. The R codes developed during the current study, are openly available in the Forge IRD repository, https://forge.ird.fr/espace-dev/personnels/teillet/bio-mod2-and-sampling-biais.git.

**Funding:** This work has been funded by the French space agency CNES, the French Occitanie Region, and the University of Montpellier. This work also received financial support from UMR Espace-Dev, IRD for a Master student's internship.

**Competing interests:** The authors have delcared that no competing interests exist.

*albopictus* larvae in public spaces, where small, organized urban objects and large patches of vegetation increase the likelihood of larval presence. This study highlights the potential of very high-resolution remote sensing and species distribution modeling for enhancing urban mosquito control strategies, ultimately contributing to improved public health policies outcomes in the face of vector-borne disease threats.

## 1. Introduction

The global expansion of the *Ae. Albopictus* (commonly known as the Asian tiger mosquito), fueled by climate change, brings new public health issues in many regions, especially in Asia and Europe, due to its role as a primary vector for diseases such as dengue, chikungunya, and Zika. Dengue fever has alone accounted for 6.5 million cases in 2023, with 7,300 deaths recorded, affecting 80 countries where the disease is endemic [1,2]. Originally native from Southeast Asia, *Ae. albopictus* has over the past few decades expanded its geographic range in response to international trade, benefiting from ideal breeding conditions during maritime and air transport of goods such as tires or ornamental plants [3,4]. In Europe, *Ae. albopictus* was first detected in Albania in 1979 and has since spread across most northern Mediterranean countries (e.g., from Spain to Greece), including Italy in 1990 and France in 2004 [5], and keeps moving northward. Its geographical expansion is also due to the capacity of its eggs to survive in temperate climates by entering diapause, a state of dormancy that allows them to survive cold winters [6,7].

In the absence of an effective vaccine against those viruses, the reduction of breeding sites is one of the most effective control strategies for *Ae. albopictus* [8]. *Ae. albopictus* breeding sites – i.e., sites where mosquitoes deposit their eggs and where larvae develop – are commonly found in various type of artificial containers [9], such as flower pots, barrels, and tires. The urban type, such as residential areas or dense informal zones, was shown to influence the availability, suitability, and abundance of these breeding sites [10]. Rajarethinam et al. (2020) have also shown a seasonal fluctuation of *Aedes*-positive containers in urban environments [11]. The reduction of breeding sites requires cooperation from the population, especially in private spaces [12]. While the majority of breeding sites are found on private property (e.g., gardens), public spaces also host numerous containers that can serve as breeding sites for *Ae. albopictus* [13]. Targeted interventions by vector control actors could effectively eliminate some of these breeding sites, more easily accessible than those located in private properties, potentially resulting in a significant reduction of mosquito populations.

Despite the presence of *Ae. albopictus* in mainland France for the past two decades, regular dengue outbreaks have not occurred there, unlike in French tropical overseas regions, such as Martinique, French Guiana, and Reunion Island [14–16]. The increasing reports of autochthonous dengue cases in southern France highlight the growing risk of local outbreaks, particularly during the summer when environmental conditions favor mosquito proliferation. Considering the rising number of imported

dengue cases and the expansion of *Ae. albopictus* to all French regions [17,18], there is a need to better understand the distribution of larval breeding sites in public spaces to enhance control efforts and reduce the risk of local transmission.

Identifying breeding sites at the scale of a city can be time-consuming and resource-intensive due to the large amount of fieldwork required [19]. To address these challenges, risk maps and models based on remote sensing can be employed to help understand the relationship between urban areas and breeding site distribution, facilitate the spatial analysis of such sites across large territories, and help target actions that could mitigate the development of *Ae. Albopictus* [20]. Remote sensing allows for extracting large-scale proxies related to climate, socio-economic, and environmental factors that can help monitor and predict mosquito breeding sites or adult density distribution or abundance [21–23]. If the influence of climate factors, such as air temperature and precipitation, on positive breeding sites is well known [11], the influence of urban landscape configuration and composition remains poorly understood [24]. Urban areas where *Ae. albopictus* can thrive can include a complex mosaic of private and public spaces, presenting challenges for mosquito control. Those areas can include diverse landscapes where the form and density of vegetation and buildings vary. Studies have shown the direct or indirect influence of land cover or land use types on the distribution or abundance of larval breeding sites [3,25–27]. Beyond the use of spectral indices in satellite image classification, some studies have also used them directly to study relationships between the environment and *Ae. albopictus* larval breeding sites [26,28,29]. Although a few studies have examined the fine-scale impact of urban landscape characteristics on *Aedes* breeding sites, more research is needed to understand how the distribution, density, fragmentation, and size of urban objects (e.g., buildings, roads, and green areas defining the urban texture) influence these sites [24,30]. For example, Teillet et al. (2024) [24] provided insights into the links between potential breeding sites' presence and urban landscapes that can help improve the design of field protocols in territories (targeting priority areas, optimization of resources). These previous studies predominantly focused on data collected in private areas, though some research also considers specific public areas such as cemeteries, and urban parks [11,31]. Data on breeding sites are most often collected on private properties because they typically represent the majority of known breeding sites [32,33]. However, it is known that public spaces also contribute to breeding sites abundance, partly due to their design and more uneven maintenance [13,34].

Additional research on the influence of urban public spaces on breeding sites and better integration of these data into control strategies are essential to improve the effectiveness of vector control programs and reduce the risk of local transmission of dengue. To address these data gaps and optimize resources, analysis of the spatial distribution of breeding sites through modeling offers a valuable approach for vector control. Species Distribution Models (SDMs) that combine presence/absence data and georeferenced environmental data can help understand the relationship between species and their environment, and predict where a species might potentially be found [35,36]. Beyond mapping potential distributions, these models are used to better explain relationships between the species and their environment [37]. Currently, a wide range of methods are available for distribution modeling, each varying in performance and features, and presence-only records are the most commonly used data for SDM [38,39]. To address these challenges and improve spatial accuracy, robustness, and reliability of predictions, combining individual models into an ensemble approach can help mitigate these issues and improve overall accuracy [40,41]. To fill the gap in understanding the distribution of larval *Ae. albopictus* in public spaces, SDMs can help identify high-risk areas for larval presence, thereby facilitating targeted interventions in urban environments. This study places particular emphasis on urban landscape factors derived from very-high resolution imagery, as these variables are not only interesting inputs for the species distribution model, but also offer an opportunity to explore and better understand the complex relationships between urban landscapes and larval presence probability in public areas.

This paper proposes a novel approach to help understand how urban landscape structure and composition affect positive breeding sites of *Ae. albopictus* in the public domain. It aims to determine to which extent landscape factors, derived from remote sensing images, can help explain the presence of positive *Aedes* breeding sites in public areas and to use these factors to predict the presence probability of such sites in the Montpellier metropolitan area, southern France, where the presence of *Ae. Albopictus* has increased over the last two decades.

## 2. Materials and methods

### 2.1. Geographical context

Montpellier, with its metropolitan area, is a major city of the south of France located in the Occitanie region, near the Mediterranean Sea (Fig 1). It has a population of around 306,000 people, reaching over 500,000 when considering its metropolitan area [42]. Montpellier benefits from a Mediterranean climate characterized by mild winters, with an average January temperature of 7.6°C, and hot and dry summers, with an average July temperature of 24.4°C. The average annual temperature is 15.5°C. In the Occitanie region, the majority of dengue cases are imported, with 51 cases reported in 2022, 212 in 2023 and 178 in 2024 [18,43–45]. However autochthonous dengue cases have also been reported, with 12 cases in 2022, 22 in 2023 and 5 in 2024 [18,43–45]. This indicates a notable presence of locally transmitted dengue despite the predominance of imported cases, highlighting the need for effective local vector control strategies to help reduce the risk of dengue outbreaks in the region.

### 2.2. Entomological data

Entomological data collected in the Montpellier metropolitan area between November 2021 and March 2022 were obtained from operational services of EID-MED (*Entente Interdépartementale pour la Démoustication du littoral Méditerranéen*). These data were collected in specific municipalities of the Montpellier metropolitan area (Saint-Clément-de-Rivière, Prades-le-Lez, Clapiers, and Castelnau-le-Lez; Fig 1) based on the operational capacities and priorities of the EID-MED services. The initial dataset includes *in-situ* observations of potential breeding sites (n = 3811) made exclusively in the public domain (e.g., streets, parking lots), such as, containers like storm drain, telecom cable chambers (i.e., concrete block used as cable pulling box) or rainwater retention basins, likely to harbor *Ae. albopictus* and *Culex pipiens* larvae.

For each record, the type of container and its positivity (i.e., presence of water only, but in an immediate urban context considered favorable for mosquito development by the vector control agents, or presence of water and larvae) were recorded. Although the winter period between November and March is not the most suitable period for larvae presence due to the temperature conditions, the entomological prospecting carried out by the vector control agents is based on an operational assumption: if a container contained stagnant water during this season and was located in an immediate urban context considered favorable for mosquito development, it was classified as positive. Indeed, according to expert knowledge, water can remain stagnant over long periods once these containers are supplied with water by rainfall or human activities (e.g., cleaning public spaces, washing cars). Telecom cable chambers do not drain and storm drains tend to be poorly maintained, causing water to accumulate. The water retained at the bottom of storm drains and telecom cable chambers also results from design problems, as observed by EID-MED agents during prospects, a challenge discussed by Carrieri et al. (2011) in Italy. Conversely, if no water is present during this period, this demonstrates that the container is sufficiently watertight or drained to prevent water accumulation under any conditions throughout the year, and it is considered as being negative for larvae (absence). Several types of public spaces breeding sites were inspected: storm drain, open retention basins, telecom cable chambers, urban pits, rural pits, underground waste garbage cans, and reservoir structures. In this study, only storm drains (Fig 1a) and telecom cable chambers were considered, as they are the structures the most likely to harbor *Ae. albopictus* larvae ([46,47], Fig 1b), and because other containers favorable to *Ae. albopictus* were too rare in the database. This entomological dataset was not derived from an exhaustive census of all containers across the city, but from exploratory surveys with an operational purpose. Therefore, 265 positive breeding sites (i.e., 8.5% of the data) were considered as presence data in the model, while 3088 negative breeding sites were considered as absence data.

While climatic parameters such as temperature and precipitation are recognized as key drivers of mosquito larval development (11,28), their inclusion was not feasible in this study due to some limitations. First, available climatic datasets have spatial resolutions that are too coarse to capture the highly localized scale of the specific

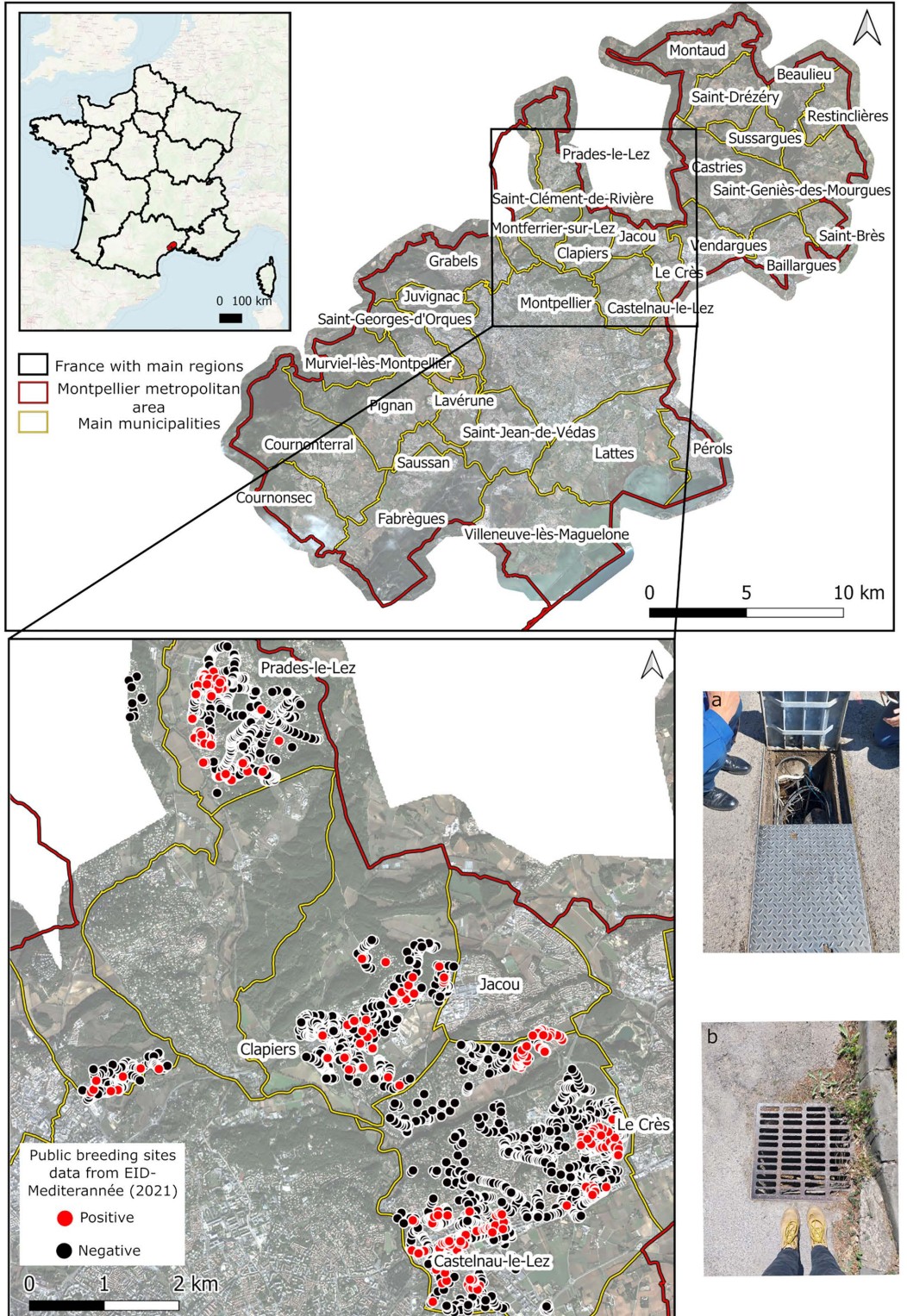

**Fig 1. Studied site and entomological data composed of positive or negative containers capable of harboring *Ae. albopictus* mosquito larvae in public spaces.** Photos of prospected breeding sites: (a) telecom cable chambers and (b) storm drain. Photo credit: Héloïse Pottier (2024).

breeding sites investigated (22,23). Additionally, based on expert knowledge from EID-MED and as demonstrated by Carrieri et al., 2011 and Hounkpe, 2012 [13,46], design issues causing persistent water retention in the specific breeding sites, combined with the fact that containers can also be filled by human action, likely reduce the influence of precipitations variability on the filling dynamics of these breeding sites. We therefore decided to exclude climatic factors to hence focus solely on urban landscape variables to evaluate the influence of the structure and composition of the immediate geographical context (urban landscape) on the positivity of these specific containers.

## 2.3. Remote sensing data

Two very high-resolution Pléiades satellite images of the study area acquired on November 2 and 6, 2021 were used in the study. Orthorectified Pléiades images include a 50 cm resolution panchromatic band and four 2 m resolution multispectral bands (blue, red, green, near-infrared). A mosaic of the two images was made using Orfeo ToolBox (OTB) software v. 8.1.2 to create a complete image of the Montpellier metropolitan area. The use of very high-resolution imagery allowed for the detailed characterization of urban landscape features relevant for studying mosquito breeding sites, including vegetation patches and built-up areas [14,15,22,24].

## 2.4. Extraction of explanatory variables

Normalized Difference Vegetation Index (NDVI, [48]) and Brightness Index (BI, [49,50]) were computed using the formula $NDVI = \frac{(NIR-RED)}{(NIR+RED)}$ and $BI = \sqrt{(RED^2 + NIR^2)}$, where "NIR" refers to the reflectance measured in the near-infrared band and "RED" corresponds to the reflectance measured in the red spectral band. We selected these specific spectral indices because they facilitated the extraction of a vegetation layer using the NDVI and abuilt-up area layer using the BI from the satellite imagery. A threshold was applied to the NDVI (> 0.2) to retain only areas with significant vegetation cover [51]. For the BI, values above 500 were selected to obtain a built-up area layer. To improve this urban layer, we used the NDVI vegetation layer to reclassify the areas classified as buildings by the BI but where there actually was vegetation. The process was realized using the QGIS software v. 3.16.

Textural indices from Pléiades were extracted using the FOTOTEX algorithm [24,52] to characterize the structure of urban types within the study area. In FOTOTEX, Fourier transforms combined with principal component analyses (PCA) convert the textural information within the image into a frequency signal and reduce it into three principal components (PC1, PC2, and PC3), representing distinct texture indices. FOTOTEX was applied to the panchromatic image (PAN) with an analysis window size of 201 pixels to identify spatial patterns that represent urban areas in their globality. Additionally, FOTOTEX was applied to spectral indices (NDVI and BI) with a window size of 101 pixels (conversion to meters depends on the spatial resolution; i.e., at 2 meters per pixel, this corresponds to 202 meters) to capture finer spatial patterns associated specifically with vegetation and urban structures [24,52]. The derived texture indices were analyzed by dividing point clouds, where each point represents an analysis window, into 12 angular sectors (or quadrants). For each quadrant, the four windows that are the furthest from the origin of the axes are identified and help represent the content of the initial image (PAN, NDVI, or BI) (see Teillet et al., 2021 [52] for details). Then, we created a colored composition of the three principal components, which allows for a rapid assessment and overview of the spatial organization. Analyses were conducted using the Python package "fototex 1.5.9" [52].

Finally, a 202 meters grid was created based on the output resolution of the textural indices calculated on the NDVI. The resampling of textural indices calculated from panchromatic images was processed using R Stats software v. 4.4.1. Statistical measures, such as mean, standard deviation, and maximum values were calculated within each grid cell over BI and NDVI with QGIS software v. 3.16. This approach allowed obtaining statistics associated with vegetation pixels (NDVI) and urban pixels (BI). Table 1 lists all variables used in this study.

**Table 1. Satellite data, method for layer creation, and variables extracted from grid cells (PAN = Panchromatic).**

| Satellite data | Band | Method for layer creation | Spectral Indices | Variables extracted from grid cells | Related variable names |
|---|---|---|---|---|---|
| Pléaides Multispectral | NIR, RED | Equations | BI | Mean of BI pixels | BI mean |
| | | | | Standard deviation of BI pixels | BI stdev |
| | | | | Maximum of BI pixels | BI max |
| | | | NDVI | Mean of NDVI pixels | NDVI mean |
| | | | | Standard deviation of NDVI pixels | NDVI stdev |
| | | | | Maximum of NDVI pixels | NDVI max |
| Pléaides Panchromatic | PAN | FOTOTEX | | 3 principal components associated with the panchromatic band | FOTO PAN PC1, FOTO PAN PC2, FOTO PAN PC3 |
| | | FOTOTEX | BI | 3 principal components associated with BI | FOTO BI PC1, FOTO BI PC2, FOTO BI PC3 |
| | | FOTOTEX | NDVI | 3 principal components associated with NDVI | FOTO NDVI PC1, FOTO NDVI PC2, FOTO NDVI PC3 |

## 2.5. Sampling bias

At the scale of the study region (i.e., the Montpellier metropolitan area) observations can have an environmental sampling bias when not all environmental conditions in the study area are sampled [53]. If some environmental conditions in the area are not represented in the samples, the model built with the samples may not accurately reflect the entire region. To assess a potential sampling bias in our study, a PCA was applied to urban landscape variables to represent study area cells in an orthonormal space, well-suited to Euclidean distance calculation and hereafter referred to as environmental space. The distance between each cell and the nearest sampled cell in the environmental space was calculated. These distances helped assess how well the observations covered the full range of environmental conditions of the study area [54]. In other words, for a given cell, the greater the distance to a sampled cell is, the less the observations (samples) are representative of the specific environmental conditions of this cell. As a result, the model will be less reliable for that cell. Spatializing these distances (i.e., representing them in the geographical space) helps identify undersampled areas.

## 2.6. Ensemble modeling

SDMs were implemented using the RStats biomod2 package v. 4.2-5-2 [55]. SDMs predict species distributions in unsampled geographical areas based on environmental variables [41]. Biomod2 allows building and evaluating individual SDMs and combining them into ensemble models [56]. This process increases the robustness and reliability of predictions by averaging the outcomes. To avoid multicollinearity between explanatory variables, pairwise Pearson correlations were calculated, using a threshold of 0.85, as recommended by Elith et al. (2010) [57]. As no correlation exceeded this threshold, all variables were retained for further analysis.

The following methods were tested in this study: artificial neural networks (ANNs), classification tree analysis (CTA), flexible discriminant analysis (FDA), generalized boosting method (GBM), generalized linear models (GLM), generalized additive models (GAM), multiple adaptive regression splines (MARS), maximum entropy (MAXNET), random forest (RF), eXtreme gradient boosting training (XGBOOST), and surface range envelope (SRE, also called BIOCLIM). Methods were applied using BioMod2's "*bigboss*" default parameters. Models were built using 10-fold cross-validations, where the dataset was divided into 10 datasets of equal sizes, each of them being used in turn to validate the model, while the remaining data were used to create the models [41]. The Area Under the Curve (AUC) of a Receiver Operating Characteristic (ROC) curve [58] and the True Skill Statistic (TSS, [59]) were calculated and averaged to assess individual models' performances [40]. Individual models' performance was used to guide the selection of models in ensembles, complemented by the

analysis of variable importance scores [41]. Several selections were tested, and we chose to retain the one that included a sufficient number of models (i.e., 6 individual models with an AUC between 0.70 and 0.75), with a satisfactory global AUC of the ensemble model (> 0.80), and where importance scores of variables were the highest in the ensemble model.

To evaluate the ensemble model, variable importance scores were computed with three permutation runs [40] where predictions are done after shuffling individual variables. A Pearson correlation score between the original and shuffled predictions is calculated and an importance score is calculated as $1 - correlation$. A high score indicates the variable being shuffled has a strong influence on the model, while a score close to 0 means the variable has little to no impact on the predictions [40]. Response curves were calculated by holding all other variables constant (mean value) while varying one, and then plotting the model's predictions against this variable. The resulting curve illustrates the species' response to that variable, helping to understand its ecological preferences and how the variable influences the model's predictions. To assess the probability of larvae presence map in new geographic areas, the coefficient of variation (CV), defined as the ratio of the standard deviation to the mean probability, is used as a measure of uncertainty. A high CV value indicates greater uncertainty in the model's predictions reflecting poor reliability in the predictions [40]. Fig 2 summarizes the modeling method used in this study.

## 3. Results

### 3.1. Landscapes variables

The Brightness Index (BI) was used as a proxy for built-up area features over the entire Montpellier metropolitan area (S1 File). Variations in values reflect different types of built-up surfaces, however, some surfaces can be mistaken for

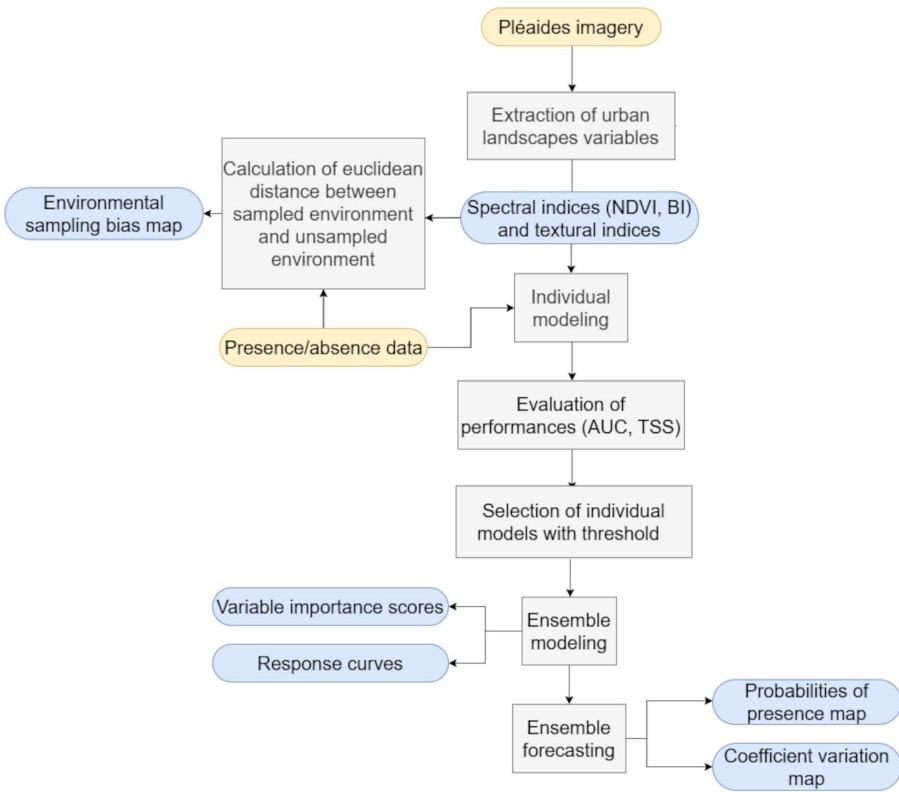

**Fig 2. Modeling method used.**

bare ground. NDVI helps identify vegetation in urban areas, where values between 0.2 and 0.8 show moderate to denser vegetation. Vegetation is sparse in the city center and in some commercial and industrial areas, but relatively abundant in certain residential areas (S1 File).

FOTOTEX, when applied on the thresholded BI, resulted in the three principal components (PC) accounting for 87% of the total variance of the image (i.e., 75%, 9% and 3% respectively). The first principal component (PC1) reveals a gradient in the density of urban objects. Windows below 0 on PC1 (e.g., windows 1 and 2 in Fig 3b) are textures composed of small, sparsely distributed urban objects with low number of repetitions (low density). As the values on PC1 increase, the textures are composed of small objects that repeat more frequently, the density of objects being higher. These textures are associated with high frequencies (i.e., high repetition of objects in the image, e.g., windows 5 and 6 in Fig 3b) resulting in a fine, fragmented, and organized pattern. Along the second principal component (PC2), a gradient in the size of urban objects emerges. This gradient ranges from small to large objects. Along PC2, we can observe small isolated urban objects (e.g., windows 1, 2 in Fig 3b) or larger isolated objects (e.g., windows 10 and 11 in Fig 3b).

The analysis of windows resulting from texture on BI offers the possibility to categorize urban structures into four different types, based on the representation of texture-based arrangements of buildings along the PC1 and PC2 axes (Fig 4). These types of urban structures evidence different levels of homogeneity and organization of urban landscapes. On the top left, large objects, sparsely distributed, follow a homogeneous and organized structure, while on the bottom left, objects are smaller and exhibit disorganization resulting in more heterogeneous landscapes. PC2 mainly gathers information related to object size. On the top right, objects are mainly large, disorganized, and follow a dense disorganized distribution (corresponding to commercial, industrial, or public infrastructures), while on the bottom right distribution, objects are smaller, still densely distributed but very organized in space (corresponding to residential urban types). PC1 mainly gathers information related to object density. Fig 4 illustrates how a textural analysis of building size and distribution (density, organization) from very high spatial resolution images like Pléiades can evidence different types of urban environments in an unsupervised manner (where no prior knowledge or complementary data are required).

By analyzing the color composition in FOTOTEX of the three main components of BI, we can highlight both the diversity and similarity of textures across the study area (Fig 3a). At the center of the map, the texture is associated with the dense city center, with little vegetation (purple color). Surrounding this central area, we observe a ring with textures varying between residential zones and mixed-use areas. Textures leaning towards green are associated with commercial areas, exhibiting a coarser texture with a variety of objects. Finally, peripheral municipalities with residential areas are shown in light pink. This phenomenon illustrates an urban gradient, ranging from a dense city center to a peri-urban periphery, where building density gradually decreases and vegetation becomes more prominent. This gradient can here be observed at different levels, Montpellier as the main city center (in purple), is surrounded by peripheral secondary centers (in pink), with a similar urban gradient structure visible around each of these secondary centers.

FOTOTEX, when applied on the thresholded NDVI, resulted in the three PC accounting for 87% of the total variance of the image (i.e., 74%, 10% and 3% respectively). PC1 is composed of windows below 0, with large single patches (windows 1 and 2; Fig 5). As values on Axis 1 increase, patches of vegetation are more fragmented and repeat more frequently (corresponding to high-frequency textures, windows 5 and 6; Fig 5). In windows 7 and 8, textures are still fragmented but vegetation patches are larger (Fig 5). Along PC2, textures are composed of large vegetation patches (windows 1 and 2) that appear to be increasingly fragmented (windows 8, 9, and 10; Fig 5b).

By analyzing the color composition of the three main components of NDVI, we observe an heterogeneity in the distribution of vegetation across urban areas (Fig 5a). Textures associated with red/pink colors (PC1) indicate a majority of small, fragmented objects that correspond to peripheral municipalities associated with residential areas with lots of vegetation. In the city center of Montpellier and peripheral areas, textures associated with green color (PC2) correspond to areas where vegetation is not predominant in fragmented small patches but in small isolated patches or larger and not fragmented patches (parks or hedges).

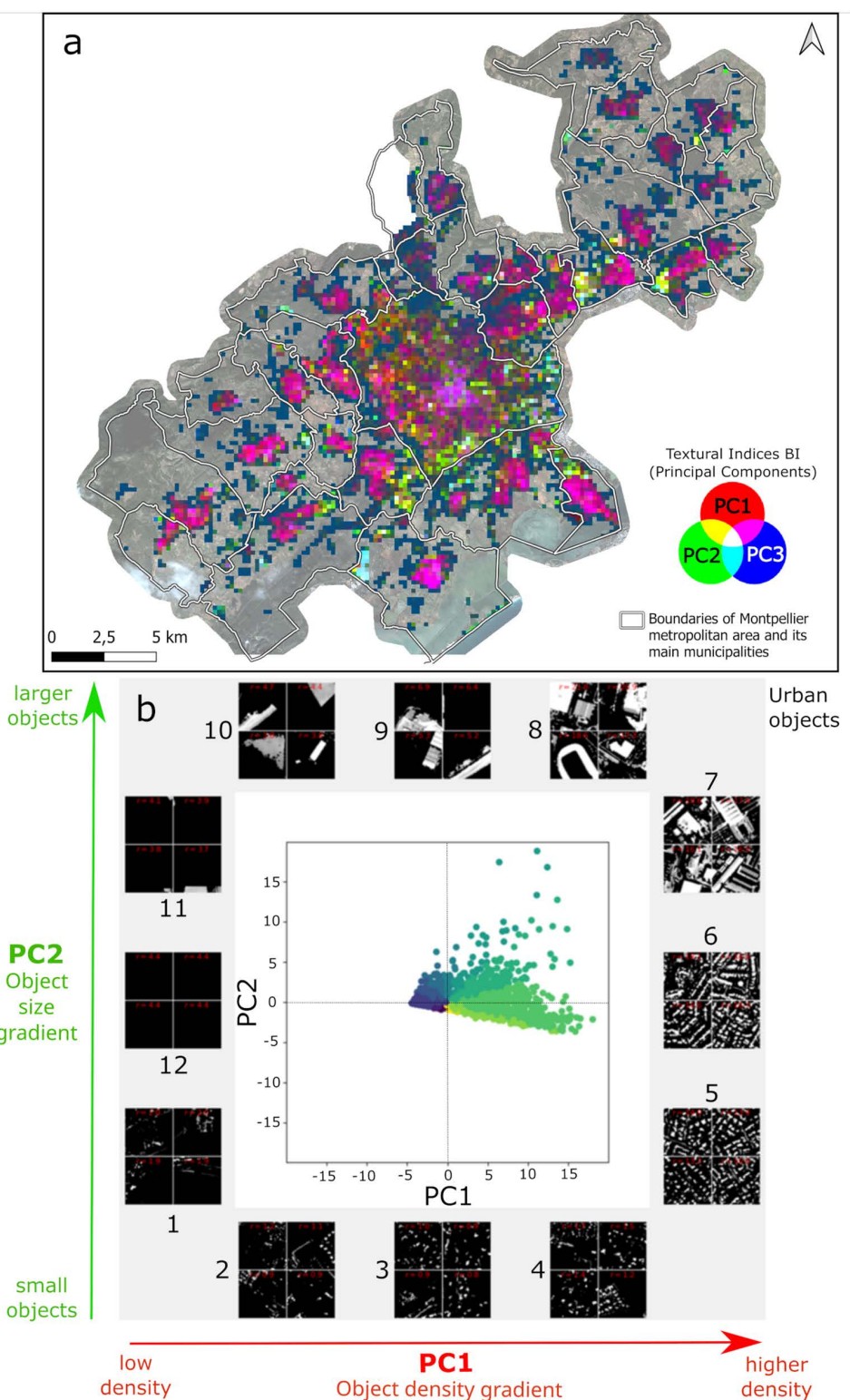

**Fig 3. Textural indices (first three principal components – PC) produced by FOTOTEX over the Brightness Index (BI).** (a) Textural indices in a red-green-blue composite image; (b) Projection of windows in the factorial plane composed of the first two PC (window size = 202 m). Colors correspond to the different angular sectors of the factorial plane. Image subsets correspond to the analysis windows furthest from the axis origin, for each angular sector. Gray areas are considered urban objects (buildings) and black corresponds to no data or vegetation.

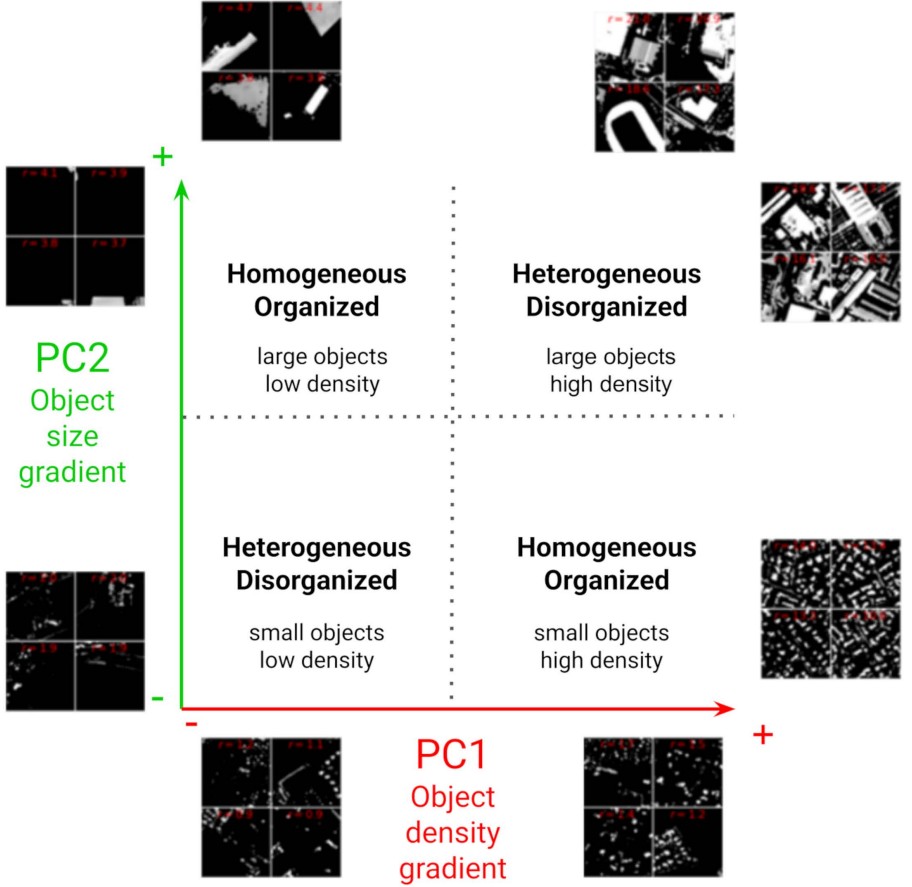

**Fig 4. Representation of texture-based structure of urban areas on the PC1-PC2 axes produced by FOTOTEX over the Brightness Index (BI) where the levels of homogeneity and organization of urban landscapes are indicated.**

In the panchromatic image, which does not distinguish between vegetation and urban objects, patterns correspond to the general structure and arrangement of the urban landscape (S2 File). Along PC1, the patterns are coarser, representing open fields with hedges, gradually transitioning to finer urban structures. On PC2, the transition moves from finer urban structures to coarser patterns characterized by large, irregularly arranged buildings.

### 3.2. Sampling bias

The distance between each cell and the nearest sampled cell in the environmental space displays a strongly positively skewed distribution with a mean value of 1.8 and first and third quartile at 1.3 and 2.6 respectively (Fig 6). Sampled environments with a distance of 0 (i.e., well sampled) are highlighted in light yellow (Fig 6a). Environments that were not sampled but are close to sampled contexts exhibit very low to low distance (up to 1.6) and are distributed throughout the area (depicted in light green in Fig 6). As the distance increases (moderate class), the environments become more distinct from the sampled areas but retains some resemblance to the sampled contexts. The corresponding textures are associated with housing neighborhoods as well as collective housing and commercial areas (Fig 6b). Higher values (above 2.0), shown in green and dark green colors, represent environments that are considerably different from the sampled ones. These include for example the city center of Montpellier, with dense buildings and fewvegetation (Fig 6c). Extremely high

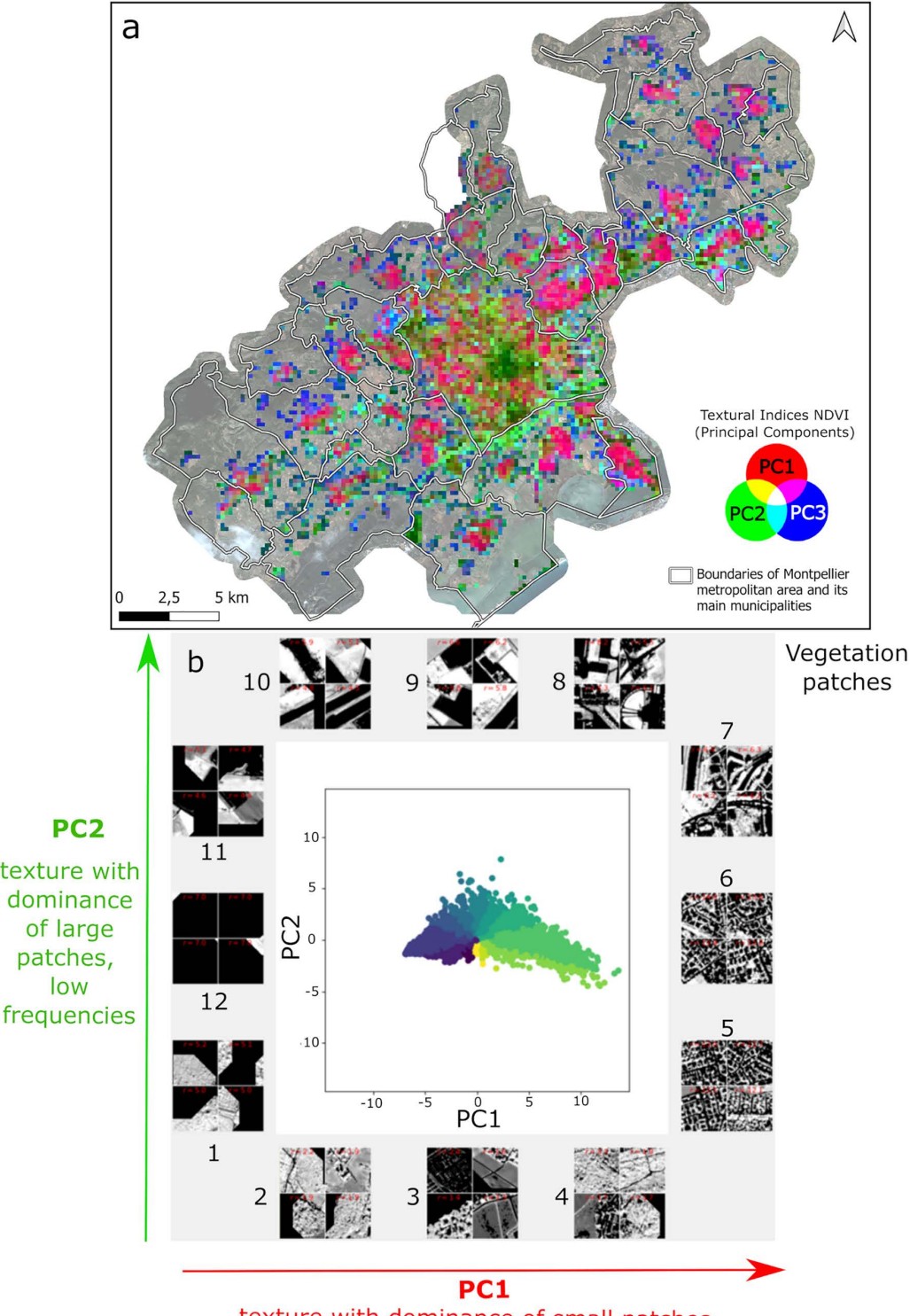

**Fig 5. Texture indices (first three principal components – PC) produced by FOTOTEX over NDVI.** (a) Textural indices in a red-green-blue composite image; (b) Projection of windows in the factorial plane composed of the first two PC (window size = 202 m). Colors correspond to the different angular sectors of point cloud individuals. Images subsets correspond to the analysis windows furthest from the axis origin, for each angular sector. Black areas correspond to non-vegetation (i.e., buildings and no-data), gray areas are vegetated areas.

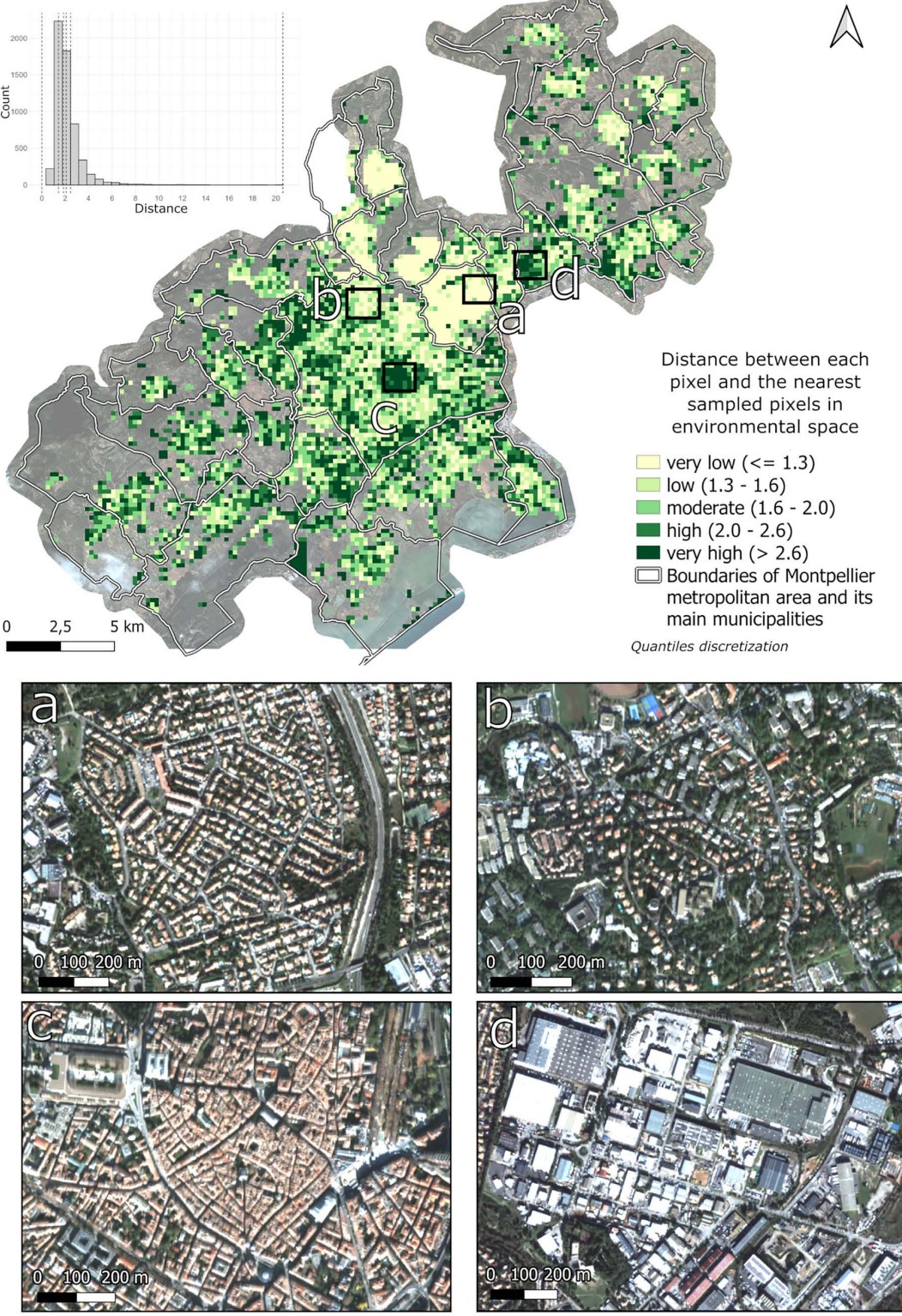

**Fig 6. Distance map between sampled pixel and non-sampled pixel in environment space with a focus on different urban landscapes.** Landscapes examples of (a) sampled environments (very low distance), (b) housing neighborhood with collective housing and commercial areas (low to moderate distance), (c) city center (high to very high distance), and (d) industrial/commercial zone with parking lots (very high distance).

values, with a max distance value of 20, correspond to highly distinct urban objects, such as quarries and parking lots, with very large complexes (Fig 6d).

### 3.3. Modeling

**Individual SDM models.** The highest performing SDMs were the CTA and GBM models, with AUC values of 0.77 and 0.78, respectively. The ANN, FDA, GLM, MARS, MAXNET, and XGBOOST models showed moderate performances, with AUC values ranging from 0.70 to 0.75. In contrast, three models – GAM, RF, and SRE – showed lower performances, with AUC values below 0.70 (Fig 7). Variable importance score of each individual model is presented in supporting information (S3 File). The standard deviations across all algorithms are relatively consistent, ranging with AUC ranging from 0.04 to 0.05 and TSS ranging from 0.08 to 0.10.

**Ensemble model.** Ensemble model selections were made based on the average AUC values of individual models (Fig 7). To balance predictive performance with an understanding of the importance of variables, we kept individual models with an AUC between 0.70 and 0.75 (i.e., ANN, FDA, GLM, MARS, MAXNET, XGBOOST), which resulted in an ensemble model AUC value of 0.81. The other two selections (individual models with AUC > 0.70 and individual models with AUC > 0.75) showed overall AUC values of 0.84 and 0.85, respectively. Variable importance scores were lower than the first selection, although the same important variables emerged across all three test selections. By analyzing the variable importance score of the selected ensemble model, NDVI mean (0.42), the first component of texture for the brightness index (FOTO BI PC1, 0.25), the first component of texture for the vegetation index (FOTO NDVI PC1, 0.24), the third component of the NDVI indices (FOTO NDVI PC3, 0.18), and the first component of the panchromatic image (FOTO PAN PC1, 0.12) are the variables contributing the most to the model (Fig 8a). Other variables have importance scores below 0.10.

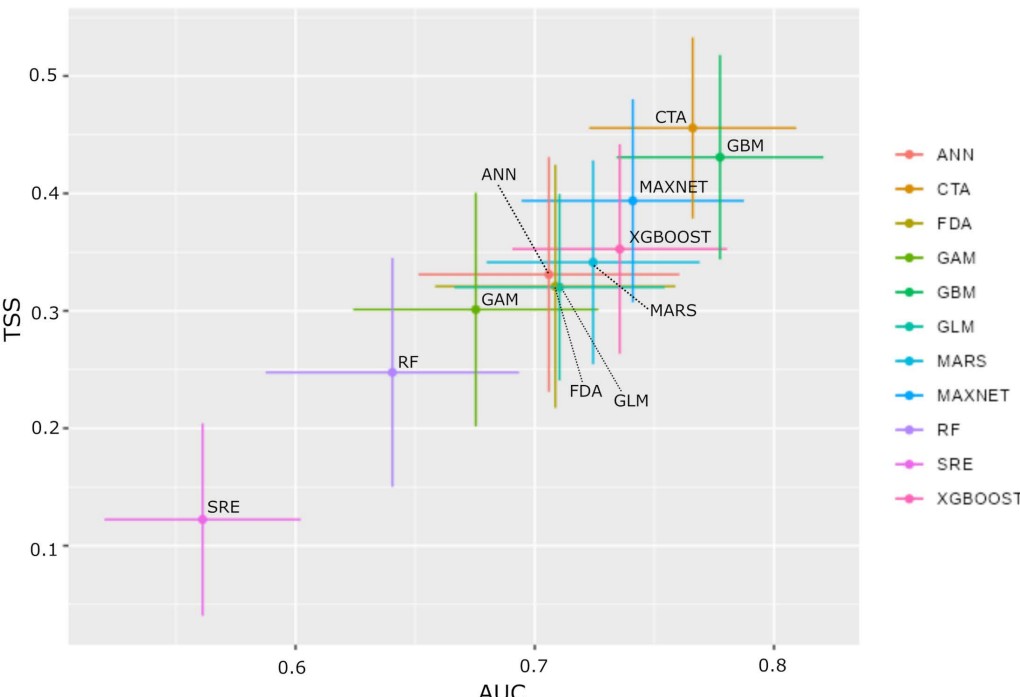

**Fig 7. Area Under the Curve (AUC) and True Skill Statistic (TSS) averaged across all model runs, along with their respective standard deviations.**

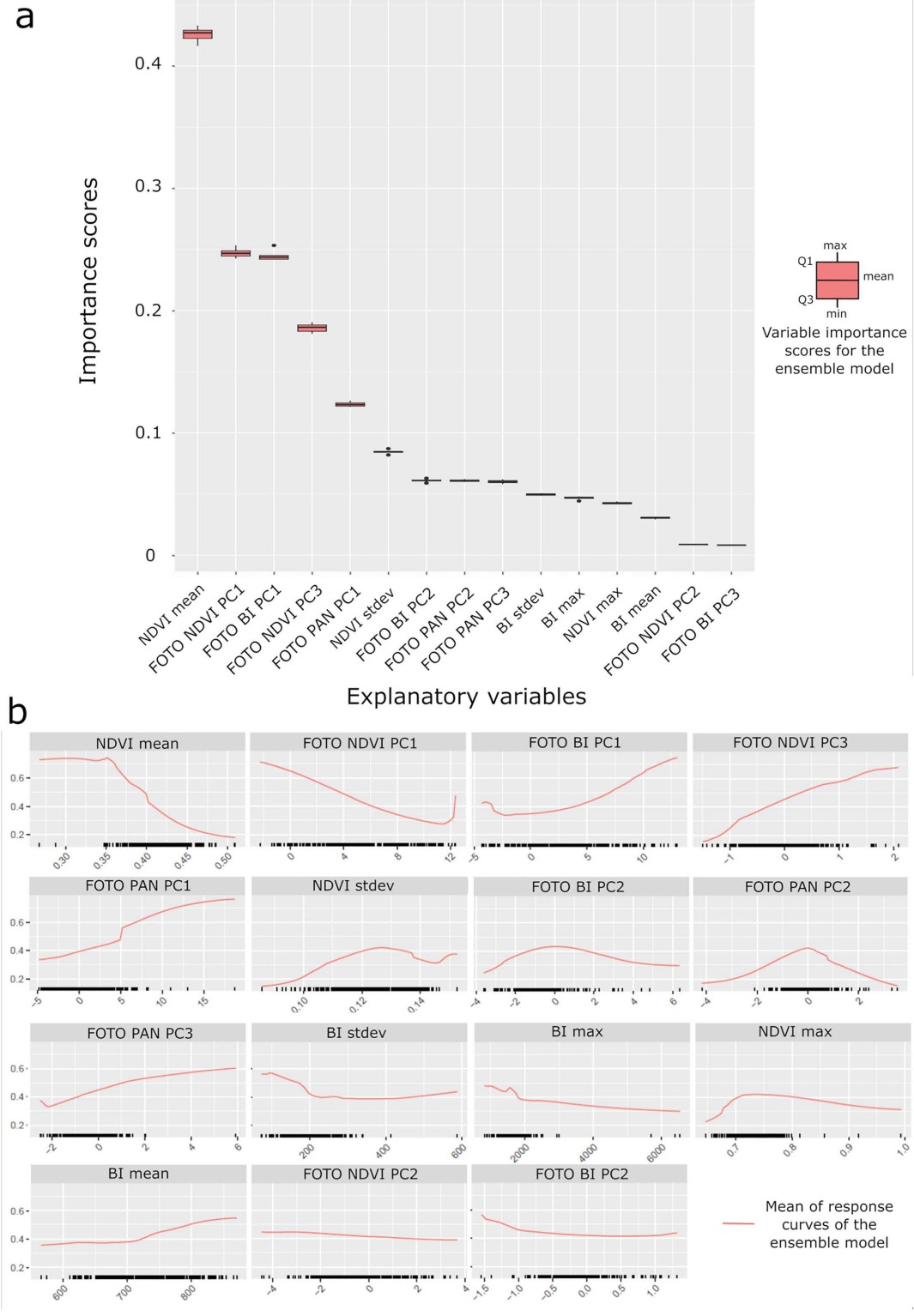

**Fig 8. Variable importance scores and response curves for the ensemble model.** (a) Variable importance scores for each explanatory variable and (b) response curves with mean as a constant for the ensemble model.

Relationships between the probability of positive breeding sites and environmental variables were analyzed using response curves (Fig 8b). The probability of a positive breeding site decreases rapidly above NDVI = 0.35. Regarding texture indices, the probability of a positive breeding sites strongly increases when the first BI component has values above 5, with probabilities exceeding 0.4. This increase is also observed when the first PAN component increases. The probability of presence also increases linearly with the values of the third NDVI component. In contrast, the probability of larval presence decreases when the first NDVI component increases. The other variables show less influence on probability presence and are fairly constant (i.e., FOTO BI PC3, FOTO NDVI PC2).

**Probabilities of larval presence.** The ensemble model was used to predict the probability of larval presence (positive public breeding sites) across the metropolitan area of Montpellier. The predicted probabilities range from 8% to 92%, with a mean of 41%. The coefficient of variation (CV) ranges from 6.6 to 170, with a mean of 51.

The highest probabilities of larval presence are mainly located in the western and northeastern parts of the metropolitan area of Montpellier (classified as 'high' to 'very high'), where uncertainty is 'low' to 'very low' (below 40, Fig 9a). In contrast, probabilities are 'very low' in the sampled area of Castelnau-le-Lez, where uncertainty is 'high' to 'very high' (above 40). Although downtown Montpellier shows 'low' to 'moderate' probabilities (0.2 to 0.6), a 'very high' uncertainty is observed in the area (>40) (Fig 9). Moreover, the map of unsampled environments reveals that these areas significantly differ from the sampled environments, with 'very high' distance values (Fig 4).

## 4. Discussion

### 4.1. General approach

This study aimed to assess how very-high resolution remote sensing images can (i) help understand the influence of urban landscapes on *Ae. albopictus* larvae breeding in drainage and telecommunications urban network infrastructures in public spaces, and (ii) predict the probability of larvae presence. This research was conducted in an urban context where dengue is not currently endemic, but represents a growing concern for public health. The originality of this study, carried out in collaboration with a local operational agency, lies in the study of *Ae. albopictus* larval habitats in public spaces, environments rarely explored. It is also to our knowledge one of the first attempts to integrate operational breeding site prospecting into larvae presence modeling [29,60]. Most studies focusing on *Aedes* mosquitoes larval habitat prediction using remote sensing, concern *Ae. aegypti* and have been conducted in tropical areas where arboviruses, particularly dengue, are endemic [15,30,61–63]. Other papers focus on *Ae. albopictus* larval habitats in dengue-endemic areas such as Thailand, Brazil, and Argentina [26,28,31]. Our study site is increasingly affected by vector-borne diseases, as the number of imported and autochthonous dengue cases continues to rise year after year [17,18].

### 4.2. Relationships between remotely-sensed landscape characterization and larvae presence

Remote sensing studies consistently highlighted the significant role that vegetation plays in the abundance of *Ae. albopictus* [26,29]. Our results confirm a strong influence of vegetation, with the mean NDVI being the most contributing variable in the models (both in individual SDMs and in the ensemble model) for predicting the presence of positive breeding sites in the public domain. Such finding was also demonstrated by Little et al. (2017) [29], where NDVI was found to be an important indicator of juvenile mosquito occurrence. However, our results suggest the relationship is more complex than it appears, as the probability of larvae presence decreases when the NDVI increases from 0.3 to 0.5 (Fig 6b). Such NDVI values typically describe sparse and moderate vegetation, which in our case would be less favorable to the presence of larvae, something in agreement with previous studies [31,64–66].

The response curves of the textural indices with NDVI reveal that the probability of larvae presence reaches a minimum when the landscape associated with vegetation exhibits a fine texture with organized and fragmented patterns (Fig 3b, 6b). These results indicate that if the vegetation is fragmented into small patches, the urban landscape is less favorable to larvae in public spaces. This observation aligns with previous studies, such as those by Reiskind et al. (2010), and Cianci

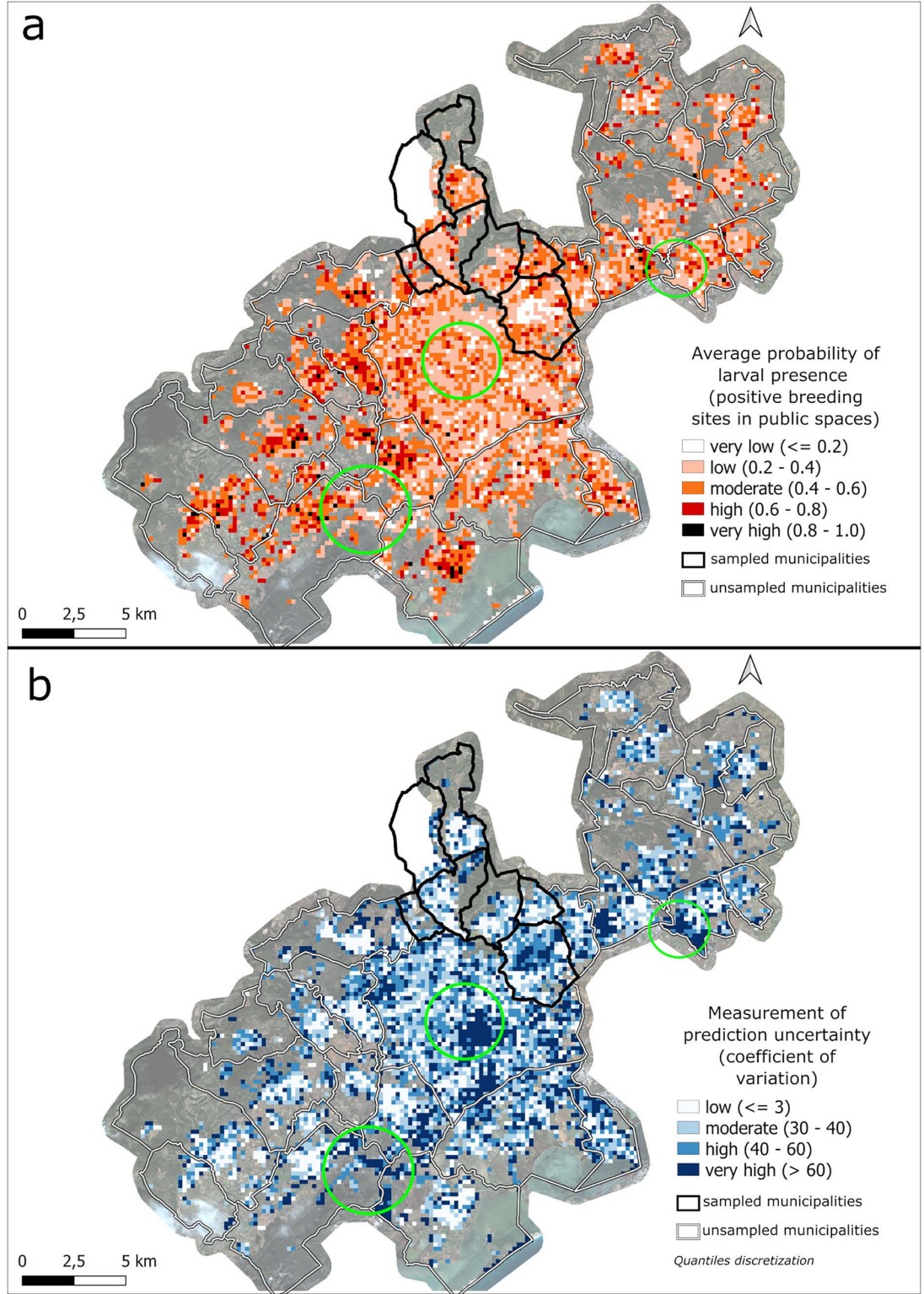

**Fig 9. Probability of presence of *Ae. albopictus* urban breeding sites in public spaces predicted by the ensemble model, with associated uncertainty (coefficient of variation) across the metropolitan area of Montpellier.** Green circles correspond to specific areas where probabilities are low to moderate but with a high level of uncertainty.

et al. (2015) [66,67], which also demonstrated that the morphology of urban vegetation can influence the presence of mosquito larvae. Such a finding could help advise urban planning strategies by providing guidance on the morphology of vegetated patches that should be integrated in urban areas, especially in public spaces, to minimize the risk of vector-borne diseases.

Textural indices obtained over BI showed their ability to discriminate urban types according to two texture gradients of density and size of objects (Fig 3b). On the one hand, we observe that the probability of presence of *Ae. albopictus* larvae in public spaces increases when the value of PC1 increases, suggesting that the presence of larvae is associated with high-frequency textures composed of dense, small urban objects. On the other hand, the probability of presence decreases when PC2 has very high values, suggesting that patterns are composed of coarse and irregularly organized urban objects. This could indicate that the presence of larvae in the public domain is more associated with highly dense, fragmented, organized small urban objects, rather than large heterogeneous urban objects. The conclusions extracted from the two variables point out that, residential areas are more prone to gather the conditions that favor larval presence.

### 4.3. Pléiades Imagery and texture analysis

Previous studies used various satellite images to characterize the urban environment (e.g., MODIS, SRTM, SPOT5, Sentinel 2, Ikonos) and associated it with *Ae. albopictus* breeding sites or adult mosquitos' presence [28–31,61]. We chose to use Pléiades imagery to extract spectral and textural indices over the entire area. The use of very high spatial resolution imagery to extract spectral indices and combine them with texture analysis allowed us to refine the studied objects and achieve a better interpretation of textures [24,68]. In addition, the use of spectral indices as FOTOTEX inputs reduce the effect of spectral variability, which can be explained by the slight difference in angle of incidence when acquiring the two images making up the mosaic image of the entire area. This effect can be seen when the FOTOTEX algorithm is applied to the panchromatic image since the radiometric information of the panchromatic band is considered in the calculation (S3 File, [52]). This effect may also explain the lower influence of the panchromatic band in ensemble modeling. However, the textural analysis approach enables the characterization of different types of urban environments in an unsupervised manner, which can be directly integrated as a continuous variable in predictive models or used to generate urban type maps.

### 4.4. Ensemble modeling and prediction of larvae presence probability

Spectral and textural indices were used to predict larvae presence probability (i.e., positive breeding sites), while earlier studies mainly used remote sensing to better understand the effect of land cover on larvae distribution or abundance [26,29,31]. Some studies have focused on predictions using data from *Ae. aegypti* breeding sites and ovitraps [61,63,69], but few have specifically addressed predictions of *Ae. albopictus* breeding sites [28]. In regions where dengue is present but not yet endemic, health concerns tend to be less critical. As a consequence, resources for systematic vector surveillance are more limited, which in turn affects the data availability and reduces the volume of data that can be used to build robust predictive models. In contrast, French overseas regions with regular dengue outbreaks [24,69,70], allocate more resources to breeding site monitoring due to greater public health concerns, dedicating more manpower on more frequent monitoring campaigns on the field. In areas where *Ae. aegypti* is the main vector and dengue is endemic; prediction and modeling can enable better management by minimizing the risk of epidemics and anticipating crisis management of an already well-established public health problem. Our study fills a gap by applying ensemble SDMs to predict *Ae. albopictus* larvae presence probabilities in public spaces, which could improve prevention and control strategies in regions where dengue is present but not yet endemic. Indeed, in such areas, those predictions have the potential to be used to assess and prevent epidemic risk, especially when populations are not immune.

Studies that model and predict larval presence probability tend to use models such as decision trees or MaxEnt [28,61,69]. Many researchers cite the superior predictive performance of ensemble modeling compared to the use of a

single model [56,71]. However, some individual predictive models, when evaluated on their own, have been shown to out-perform ensemble models that combine multiple predictions [72]. In our study, individual models were constructed using the "*bigboss*" parametrization proposed by the biomod2 team to benefit from a parametrization suitable to each model. The performance of individual models varied due to their diverse capabilities to handle linear versus non-linear relation-ships – such as Random Forest and Generalized Linear Models – or their distinct mathematical principles, like MaxEnt or Artificial Neural Networks (ANNs). All the individual models delivered satisfactory and fairly consistent performances, lead-ing to the development of a consistent and robust ensemble model. By combining the six individual SDMs, we achieved an overall improvement in the AUC, surpassing the AUC of individual SDMs. Cross-validation of each model considered separately provides an indication of model quality, thus contributing to the evaluation of the ensemble model and its per-formance, but could be further strengthened by validation with completely independent data to assess its reliability [41,56].

Probability maps identified urban environments favorable to *Ae. albopictus* larvae presence in the drainage and tele-com urban network infrastructures. However, these results should be treated with caution in areas where the uncertainty of the overall model is high. Sampling biases map could be used to readjust sampling protocols by prioritizing unsampled urban contexts to ensure that the diversity of environments is well represented. Despite these uncertainties and the need to improve protocols, probability mapping can be used to target areas that could be prioritized for enhanced maintenance and cleaning of urban breeding sites like storm drains and telecom cable chambers to avoid water accumulation over the year. Therefore, our results help characterize the environment favorable to the presence of positive breeding sites. Recent studies carried out on *Ae. albopictus* in Montpellier can help us estimate the density of mosquitoes or larvae using a mechanistic model based on meteorological data, environmental variables, and potential breeding sites data, focusing on private areas and their surroundings, without explicitly considering public spaces [70]. Our results on larvae presence probability can provide complementary information on *Ae. albopictus* and could be used as inputs or contribute to improv-ing mosquito density modeling by providing more precise information on positive breeding sites. However, it is important to acknowledge that larval presence alone does not fully represent vector competence and transmission risk, which depend on additional biological, environmental and socio-economic factors, including population vulnerability or human behaviors.

### 4.5. Dataset evaluation and sampling bias

The EID-MED field agents collected the entomological data used in this study to identify which breeding sites in pub-lic areas across neighborhoods of the metropolitan area of Montpellier are likely to be positive. Breeding sites in public areas are rarely considered in modeling approaches [28,63,73]. However, recent studies like Haddawy et al., 2019 have explored the use of Google Street View imagery to identify breeding containers in public spaces [74]. This raises ques-tions about the resources dedicated to data collection in the field. Current resources in Montpellier metropolitan area only allow for a limited number of prospections in few environmental contexts, while a much wider sampling both in private and public spaces would be required to improve knowledge on larvae distribution and abundance. Such information would be key to have realistic data, to obtain more accurate model outputs, and to better inform vector control strategies.

An unusual aspect of the entomological dataset used is that data on breeding sites and their positivity to larvae were collected in the metropolitan area during the winter period (i.e., November to March), which is not the period during which larvae are more likely to be found due to predominantly unfavorable temperature conditions. However, due to relatively mild winters in the region, favorable conditions are not excluded in specific places and times during this period. These aspects of the dataset are mainly explained by the fact that EID-MED primary mission is the surveillance of other mos-quito species in the coastal wetlands of the entire Mediterranean area [75]. All agents are fully mobilized to this task between April and October, leaving no opportunity for data collection in urban areas during this period. The expertise of field agents reveals that breeding sites with poor design or insufficient maintenance tend to retain stagnant water during periods of unfavorable conditions (November to March), these sites will stay in water throughout these months and will therefore contain larvae during the period favorable to *Ae. albopictus* mosquitoes (April to October). September and

October are the most rainy months of the year in the Montpellier area, characterized by strong localized rainy events with high accumulation of precipitation [76]. Public breeding sites that drain well will not fill up with water during those two months and will not retain water. However, if there is a design flaw in some public breeding sites, water that will fill them up will partially stagnate and remain present throughout the winter if no manual drainage is made, since the weather conditions will prevent evaporation. The presence of stagnant water inside breeding sites underlines that precipitation during the days preceding the fieldwork is not a necessary driver of, and is not correlated to breeding site watering and larvae appearance dynamics. These assumptions rely on expert knowledge and could benefit from validation through comparative data collected during the summer. It would be highly valuable to monitor the positivity of these "winter potential breeding sites" during summer to better evaluate the reliability and representativeness of the winter dataset. In addition, another source of water should also be considered in such studies since the filling of these types of breeding sites can also be linked to human activities, such as street cleaning and maintenance of rainwater infrastructure. That is why meteorological variables like rainfall are not considered in the models.

The entomological data provided were not uniformly distributed across the Montpellier metropolitan area, with samples taken only in the northeastern of the area. This resulted in an uneven sampling of the different environmental contexts encountered in the study area (Fig 4), being limited to certain types of urban environments. To improve the quality of this sampling and thus reduce environmental bias, it would be necessary to sample in more diversified environmental contexts [33]. The spatialized indicator presented in Fig 4, based on the distance between non-sampled and sample sites in the environmental variable space, could help improve the sampling protocol by prioritizing locations corresponding to unsampled environments (those exhibiting high indicator values). This mapping would therefore make it easy to identify the areas to be sampled as a priority by EID-MED agents, to improve the quality and representativeness of sampling.

Finally, our dataset contains only 8.5% of positive breeding sites, which represents far less presence data than absence data. Moudrý et al. (2024) suggest that the size of our dataset and the uneven distribution between presence and absence data allow us to carry out robust SDM [77]. However, the performance of SDMs can generally be limited, even with a large sample size, as it does not guarantee the complete representation of the large distribution area (ecological niche) of *Ae. albopictus* [77,78]. A more balanced sample distribution could still improve the performance of the model [77] and a sampling bias correction method could be used when this is low [54]. More broadly, the specific context of emerging arboviral risks in this region means that entomological surveillance protocols have traditionally been less intensive than in endemic tropical areas, which partly explains current limitations in data coverage and sampling diversity. This context of emerging risk in a temperate region adds a particularly valuable perspective to this study, as it reflects the challenges encountered by many countries or regions now facing the emergence of these diseases. Importantly, this study highlights the relevance and feasibility of developing spatial models using data constrained by operational realities and specific contexts. Understanding these operational and logistical constraints is essential to interpreting the limitations of current datasets and to improving surveillance and control strategies. It is thus necessary to continue implementing more efficient, and optimized vector surveillance and control strategies. These strategies could benefit from stronger links between operational and research sectors, particularly in order to question the transferability of knowledge, data, methods (e.g., on sampling design, landscape characterization, data analysis and modeling approaches) related to dengue-endemic areas.

## 5. Conclusions

This study used remote sensing imagery to investigate the influence of urban landscapes on the presence of *Ae. albopictus* larvae within public spaces of the Montpellier metropolitan area, France. Spectral and textural indices derived from very high-resolution imagery were used to predict larvae presence, providing insights into the mosquito's ecology. The originality of this study is twofold: it lies both in its focus on an area where dengue is not yet endemic, and in its focus on breeding sites located in the public spaces (vs. private spaces), where the establishment of the dengue vector *Ae. albopictus* and climate change raise significant public health concerns for the years to come. The study highlighted a

role of urban vegetation on larval presence within telecom cable chambers and storm drains in public spaces, indicating however that the probability of larval presence tends to decreases with increasing vegetation index values. The analysis revealed that urban landscapes, expressed in this study as a combination of vegetation and urban objects, play a role in determining the presence of *Ae. albopictus* breeding sites in the public domain, where small organized urban objects and large patches of vegetation could increase the likelihood of larval presence. Our study of breeding sites in the public domain provides some insights into an understanding of their relationship with urban landscapes at a fine scale. Ensemble models significantly improved the prediction of breeding site presence, highlighting the importance of vegetation and textural indices. Our study highlights the potential presented by the combination of remote sensing imagery and SDMs to target and prioritize mosquito control interventions in urban areas. These predictive maps can be useful for optimizing the allocation of resources, ensuring that vector control efforts are concentrated in the most critical areas to reduce mosquito density and thus limit the spread of vector-borne diseases. In addition, by identifying areas where water is likely to accumulate, such as poorly designed or maintained public infrastructures, urban maintenance teams can take preemptive actions to eliminate potential breeding sites. This approach contributes to reducing mosquito populations and mitigating the public health risks associated with vector-borne diseases like dengue. While this study focuses on breeding sites in public spaces, research would greatly benefit from using datasets combining public and private breeding sites that would provide a more comprehensive understanding of the urban environment's capacity to sustain *Ae. Albopictus* larvae. Close collaboration between scientists and operational staff could help enhance vector control strategies and improve field actions, where an urgent response to vector-borne public health issues is needed.

## Supporting information

**S1 File. Brightness Index (BI) and Normalized Difference Vegetation Index (NDVI) calculated with the multispectral Pléiades image over the metropolitan area of Montpellier.** The top-left inset shows the dense city center of Montpellier, while the top-right one shows a residential area in a nearby municipality.
(TIFF)

**S2 File. Projection of PAN windows in the factorial plane made up of the first two principal components for a window size of 202 m.** The colors correspond to the different angular sectors of point cloud individuals. Images subsets correspond to the analysis windows furthest from the axis origin, for each angular sector.
(TIFF)

**S3 File. Variable importance score of each individual model.**
(TIFF)

## Acknowledgments

Thanks to the "EID-MED" for providing access to their entomological dataset. We also thank the '*Dispositif Institutionnel National d'Accès Mutualisé en Imagerie Satellitaire*' (DINAMIS) and the '*Institut national de l'information géographique et forestière*' (French national geographic institute – IGN) for providing the orthorectified Pléiades CNES 2022 Distribution AIRBUS DS.

## Author contributions

**Conceptualization:** Claire Teillet, Héloïse Pottier, Rodolphe Devillers, Emmanuel Roux.

**Data curation:** Claire Teillet, Héloïse Pottier, Alexandre Kerr, Frederic Jean, Gregory L'Ambert, Nicolas LeDoeuff.

**Formal analysis:** Claire Teillet, Héloïse Pottier.

**Investigation:** Alexandre Kerr, Frederic Jean, Gregory L'Ambert, Nicolas LeDoeuff.

**Methodology:** Claire Teillet, Héloïse Pottier, Rodolphe Devillers, Alexandre Defossez, Thibault Catry, Emmanuel Roux.

**Project administration:** Rodolphe Devillers, Thibault Catry, Emmanuel Roux.

**Resources:** Rodolphe Devillers, Emmanuel Roux.

**Software:** Claire Teillet, Alexandre Defossez.

**Supervision:** Claire Teillet, Rodolphe Devillers, Emmanuel Roux.

**Validation:** Claire Teillet, Rodolphe Devillers, Emmanuel Roux.

**Visualization:** Claire Teillet.

**Writing – original draft:** Claire Teillet.

**Writing – review & editing:** Claire Teillet, Héloïse Pottier, Rodolphe Devillers, Alexandre Defossez, Thibault Catry, Alexandre Kerr, Frederic Jean, Gregory L'Ambert, Nicolas LeDoeuff, Emmanuel Roux.

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
