## [Decision Letter · Decision Letter 0]

7 May 2025

Dear Dr. Teillet,

Thank you for submitting your manuscript to PLOS ONE. After careful consideration, we feel that it has merit but does not fully meet PLOS ONE’s publication criteria as it currently stands. Therefore, we invite you to submit a revised version of the manuscript that addresses the points raised during the review process.

1) Authors must focus on determining larval presence probability. 

2) Authors must develop a model that accurately predicts mosquito larval presence.

3) Please explain better the results of positive containers with water and ponds with larvae.

4) It is also recommended that temperature data be incorporated into the models.

We look forward to receiving your revised manuscript.

Kind regards,

Humberto Lanz-Mendoza

Academic Editor

PLOS ONE

Journal Requirements:

“This work has been funded by the French space agency CNES, the French Occitanie Region, and the University of Montpellier. This work also received financial support from UMR Espace-Dev, IRD for a Master student’s internship.”

3. Please expand the acronym “CNES, UMR and IRD” (as indicated in your financial disclosure) so that it states the name of your funders in full.

4.  Please ensure that you refer to Figure 2 in your text as, if accepted, production will need this reference to link the reader to the figure.

5. We note that Figure 1, 3, 5, 6, 9 and S1 S2 files in your submission contain [map/satellite] images which may be copyrighted. All PLOS content is published under the Creative Commons Attribution License (CC BY 4.0), which means that the manuscript, images, and Supporting Information files will be freely available online, and any third party is permitted to access, download, copy, distribute, and use these materials in any way, even commercially, with proper attribution. For these reasons, we cannot publish previously copyrighted maps or satellite images created using proprietary data, such as Google software (Google Maps, Street View, and Earth). For more information, see our copyright guidelines: http://journals.plos.org/plosone/s/licenses-and-copyright.

A. You may seek permission from the original copyright holder of Figure 1, 3, 5, 6, 9 and S1 S2 files to publish the content specifically under the CC BY 4.0 license. 

B. If you are unable to obtain permission from the original copyright holder to publish these figures under the CC BY 4.0 license or if the copyright holder’s requirements are incompatible with the CC BY 4.0 license, please either i) remove the figure or ii) supply a replacement figure that complies with the CC BY 4.0 license. Please check copyright information on all replacement figures and update the figure caption with source information. If applicable, please specify in the figure caption text when a figure is similar but not identical to the original image and is therefore for illustrative purposes only.

**Additional Editor Comments:**

Please provide more information on:

1) Authors must focus on determining larval presence probability.

2) Authors must develop a model that accurately predicts mosquito larval presence.

3) Please explain better the results of positive containers with water and ponds with larvae.

4) It is also recommended that temperature data be incorporated into the models.

Reviewers' comments:

Reviewer's Responses to Questions

**Comments to the Author**

1. Is the manuscript technically sound, and do the data support the conclusions?

Reviewer #1: Yes

Reviewer #2: Yes

2. Has the statistical analysis been performed appropriately and rigorously?

Reviewer #1: Yes

Reviewer #2: Yes

3. Have the authors made all data underlying the findings in their manuscript fully available?

Reviewer #1: Yes

Reviewer #2: Yes

4. Is the manuscript presented in an intelligible fashion and written in standard English?

Reviewer #1: Yes

Reviewer #2: No

Reviewer #1: The article titled "Predicting Ae. albopictus Larval Presence Probability in Public Spaces Using Very High Resolution Satellite Imagery" by Claire Teillet et al. examines the influence of urban landscapes on the presence and distribution of Ae. albopictus larvae, particularly within two specific niches.

The study evaluates environmental bias by analyzing the representativeness of sampled reproduction sites across various urban landscapes. The authors employ Species Distribution Models (SDMs) to predict larval presence in the study area, identifying the Normalized Difference Vegetation Index (NDVI) as the most influential variable, with large vegetation patches increasing the likelihood of larval presence. This research is significant due to the rising incidence of dengue cases in non-endemic regions, such as the study area.

Aerial imagery has previously been utilized as a tool for monitoring and predicting larval and adult mosquito development sites. The study's relevance lies in applying these tools in areas without sustained dengue endemicity, specifically focusing on public mosquito development sites. However, the article's emphasis on image processing and model development somewhat overshadows the primary objective of determining larval presence probability, which is only addressed between pages 432-449 in the results section. It is recommended that the authors reduce the specialized content and concentrate on developing a model that accurately predicts mosquito larval presence.

Additionally, the authors should provide more comprehensive data, as the entomological data used to monitor storm drains and telecom cable chambers lacks clarity regarding larval positivity, which is a crucial inclusion criterion. It is assumed that a census of such containers exists throughout the city, but it is unclear if this correlates with the risk areas depicted in Figure 9. Furthermore, the resolution of the satellite images should be clarified to determine if these potential larval niches and their relationship with surrounding vegetation can be effectively monitored. It is also important to ascertain whether these sites are relevant for the development of Aedes albopictus or other mosquito species, such as Culex spp., which could be correlated with the positivity index and presence of these species in the deposits.

The results of the monitoring remain unclear to me, particularly regarding the presence of positive containers with water and ponds with larvae. These data are crucial as they directly indicate the model's success and its correlation with positive deposits of mosquito larvae. How frequently have such positive containers for Aedes albopictus larvae been identified? Furthermore, it is noted that other parameters, such as temperature, have not been considered. However, the authors themselves mention in lines 585-588 that "An unusual aspect of the entomological dataset used is that data on breeding sites and their positivity to larvae were collected in the metropolitan area during the winter period (i.e., November to March), which is not the period during which larvae are more likely to be found due to predominantly unfavorable temperature conditions." Therefore, incorporating temperature data into the models may be advisable. Although I understand that MaxEnt utilizes temperature data, it appears that the authors have neither collected nor mentioned such data. Climate parameters, such as temperature, as the authors have indicated, facilitate larval growth. I hypothesize that these types of containers are located in various areas with specific microclimatic conditions; can this be documented or correlated with larval growth?

I contend that the rationale for excluding climate data is insufficiently substantiated, and I suggest that the authors emphasize or describe climatic conditions that could correlate with the likelihood of finding larvae.

In the conclusions, between lines 654-656, it is stated that "These predictive maps can be useful for optimizing the allocation of resources, ensuring that vector control efforts are concentrated in the most critical areas to reduce mosquito density and thus limit the spread of vector-borne diseases." This prompts me to inquire about the current distribution of the species in the study area. While there are cases of native dengue, the known distribution of Aedes albopictus has not been addressed. If this data were available, we could discuss the current distribution and the potential implications of other risky niches.

Additionally, it is important to note that vector density is not necessarily indicative of vector competence, and this should also be included in the discussion.

Reviewer #2: REVISION DE L ARTICLE:

Predicting Ae. albopictus larval presence probability in public spaces using very high- resolution satellite imagery

Minor concerns

Abstract:

“particularly in storm drains and telecom cable chambers in Montpellier, France. ·: Could you rephrase this sentence?

“evaluated “

What was evaluated ? Do you mean “their accuracy was evaluated”?

NVDI meaning: define the abbreviation in its first appearance in the text on line 35. Normalized Difference Vegetation Index?

42 ultimately contributing to improved public health outcomes in the face of

43 vector-borne disease threats.

Would write public health policies outcomes.

51. Ae. Albopictus ………..role as a primary vector for diseases such as dengue, chikungunya, and Zika.

Would be relevant to mention that this is the case in Asia, not so much in America.

63. Ae. albopictus breeding sites - i.e., sites where

64. 65 mosquitoes lay eggs and where larvae develop - are known to be common in a wide range of artificial containers [9]

Rephrase.

76. Despite a presence of Ae. albopictus for two decades, mainland France does not experience regular dengue outbreaks,

77 unlike French tropical overseas regions, such as Martinique, French Guiana, and

78 Reunion Island [14–16]

The phrase is confusing

Line 80 “Coupled with” “considering the” would be better suited

Line 107 “in the private domain “rather “private properties”

108 they typically represent the majority of known breeding sites [32,33]. However,

109 it is known that public spaces also account for the most breeding sites,

Contradictory sentence: the majority and the most are the same thing

Line 115 ¨analyzing¨ to be replaced by ¨analysis of the¨

Line 117

between species and their environment, and predict where a species might potentially be found [35,36].

A prediction is probabilistic hence always with a grade of uncertainty.

Line 122 “enhance predictions” what aspect of the prediction would be enhanced?

Line 124 “SDMs can help to identify” rather “can help identify”

Line 129 “ in which extent “-> to which extent

Line 139 of 306,000 People

Line 144 However, there have also been autochthonous dengue cases TO However, autochthonous dengue cases have also been reported

Line 155. That is -> such as

Line 163 “then water and larvae “. Since the precondition is that there is presence of water, the only deduction is the presence of larvae.

Line 167 “and water does not drain properly from storm drains that tend to be poorly maintained. “ rather: storm drains tend to be poorly maintained impeaching the water to drain properly.

LINE 170 “On the contrary “ rather use “ conversely”.

LINE 172-175

Several types of breeding sites were inspected in public spaces: storm drain, open retention basins, telecom cable chambers, urban pits, rural pits, underground waste garbage cans, and reservoir structures. In this study, only storm drains (Fig. 1a)

Change to

Several types of public spaces breeding sites were inspected

Give the references for this asumption

LINE 179

We hence decided to exclude climatic TO Hence, we decided

Line 189 study area -> studied area

Line 217 Then, we created a colored composition of the

Coma

Line 241

“In other words, for a given cell, the greater the distance to a sampled cell is, the less the observations (samples) are representative of the specific environmental conditions of this cell, and the less a model built with the observations will be reliable.

The sentence should be split for clarity.

In short, machine learning allows the data to dictate the form of the model, whereas conventional statistics attempts to fit the data to an investigator-specified model.

Line 212

Window size of 101 pixels (i.e 202 meters) ..Precise the conversion, pixels are superficy unit unless aligned.

Table 1: broadening of the first column to avoid dividing panchromati C.

Line 293

The city center has less vegetation, as do certain commercial and industrial zones

¿is it “in contrast with” instead of “as do”

Line 320

(where no prior knowledge or complementary data is required)

Would rather use where no prior knowledge or complementary data are required.

Line 331

Montpellier being the main city center (purple), surrounded by peripheral secondary centers (pink), and the same type of urban gradients observed for each urban center

Sentence structure unclear

Fig. 3 clarify the figure legend and improve the letters readability of scales in the figure.

Line

Line 343 Fig. 4 caption: evidenced. Indicated could be more appropriated.

Line 426 “Regarding texture indices, the probability strongly increases when the first BI

426 component has values above 5, reaching values above 0.6” Unclear: what probability must be above 0.6?

Line 488 “inform”, would be “form” or shape more appropriate?

Line 506 “in relationship with Ae. albopictus breeding sites” would “ and associate it with Ae. albopictus breeding sites” be better?

Line 528 “In regions where dengue is present but not yet endemic, health concerns are less critical, leading to more limited resources for systematic vector surveillance, affecting in turn data availability and the volume of data needed to build robust predictive models.” Phrase should be split.

Line 531 would put frequent instead of ·regular”

Line 563 would write “therefor Our results help characterize the environment favorable to the presenceof positive breeding sites.” Instead of Our results thus help characterize the environment favorable to the presence of positive breeding sites.

Line 578 would write “This raises questions about the resources dedicated to data collection in the field” instead of This raises questions of the resources dedicated to data collection in the field

Line 594 that if water remains stagnant in poorly designed or poorly maintained breeding sites

Breeding sites or stagnant water urban structure? Otherwise would sound as the city do design and maintain breeding site.

Line 599 “Public breeding sites that drain well will not fill up with water during those two months and will not retain water, but, if there is a design flaw in some public breeding sites, water that will fill them up will partially stagnate and remain present throughout the winter if no manual drainage is made, since the weather conditions will prevent evaporation.”

Split the phrase

Line 630 “It is thus necessary to continue implementing more efficient, and optimized vector surveillance and control strategies which could benefit from stronger links between operational and research sectors, particularly in order to question the transferability of knowledge, data, methods (e.g. on sampling design, landscape characterization, data analysis and modeling approaches) related to dengue-endemic areas.”

Phrase should be split

**Do you want your identity to be public for this peer review?** For information about this choice, including consent withdrawal, please see our Privacy Policy

Reviewer #1: No

Reviewer #2: No

---

## [Author Response · Author response to Decision Letter 1]

12 Aug 2025

We thank the editor and reviewers for their comments. We have uploaded a separate file entitled “Response to Reviewers”, as requested by PLOS one, to respond to these comments.

---

## [Decision Letter · Decision Letter 1]

16 Oct 2025

Characterizing urban landscapes using very-high resolution satellite imagery to predict Ae. albopictus larval presence probability in public spaces

PONE-D-25-16377R1

Dear Dr. Teillet,

We’re pleased to inform you that your manuscript has been judged scientifically suitable for publication and will be formally accepted for publication once it meets all outstanding technical requirements.

Kind regards,

Humberto Lanz-Mendoza

Academic Editor

PLOS ONE

Additional Editor Comments (optional):

None

Reviewers' comments:

Reviewer's Responses to Questions

**Comments to the Author**

Reviewer #1: All comments have been addressed

Reviewer #2: All comments have been addressed

2. Is the manuscript technically sound, and do the data support the conclusions?

Reviewer #1: Yes

Reviewer #2: Yes

3. Has the statistical analysis been performed appropriately and rigorously?

Reviewer #1: Yes

Reviewer #2: Yes

4. Have the authors made all data underlying the findings in their manuscript fully available?

Reviewer #1: Yes

Reviewer #2: Yes

5. Is the manuscript presented in an intelligible fashion and written in standard English?

Reviewer #1: Yes

Reviewer #2: (No Response)

Reviewer #1: In my view, the authors have successfully addressed the previously posed questions by refining the study's objective, as reflected in the updated title. They have highlighted key elements of their research, such as the justification for excluding climate data, and have clearly outlined the scope, limitations, and probabilistic aims of identifying sites favorable for Aedes albopictus mosquitoes. Additionally, their approach to surveillance systems is commendably balanced, marking a positive advancement. The authors stress the importance of this research in areas with emerging cases and outbreaks, though it may not be feasible in tropical regions. I recommend that researchers consider adding an index of abbreviations to enhance accessibility for the general public, ensuring that terms like NDVI, BI, SDMs, EID-MED, and OTB are explicitly defined. I am of the opinion that this work warrants consideration for publication.

Reviewer #2: Re-Revision of Predicting Ae. albopictus larval presence probability in public spaces using very high- resolution satellite imagery

The Reviewer consider the article suitable for publication in pLOS One, having considered the modification made by the authors.

Title:

The updated title corresponds to the work content.

From

Predicting Ae. albopictus larval presence probability in public spaces using very high- resolution satellite imagery

To

1. Characterizing urban landscapes using very-high

2. 2 resolution satellite imagery to predict Ae. albopictus

3. 3 larval presence probability in public spaces

The abbreviation NDVI is now described in the abstract.

The subject of the evaluation is defined satisfactorily in the abstract.

Public health policies outcomes was added.

Clarification for the geographical localization of Aedes albopictus added in the text satisfactorily.

Rephrasing of the A. albopictus breeding site made the sentence straightforward.

The minor details in lines 80,107 were attended

Rephrasing of line 109 clarify the matter.

The explanation of the ·enhanced prediction· is adequate.

The sentence “mitigate these issues and improve overall accuracy” clarifies the matter.

The modifications of line 134, 144, 149, 155 are adequate to the subject matter.

The clarification about the conditions of larval growth/presence are adequate.

Modification of the line 173 is adequate (Telecom cable chambers do not drain and storm

173 drains tend to be poorly maintained, causing water to accumulate.).

Line 170 was modified adequately.

Lines 172-175, 179,189, 241 and 217 were modified adequately.

The precision about the length units (line 205) were adequate.

The table 1 was altered correctly.

The revisor understand the clarification supplied in the revision for line 293.

The minor corrections of lines 320, 331, Fig.3, 343, 488,506, 528, 531 and 563 were modified satisfactorily.

**Do you want your identity to be public for this peer review?** For information about this choice, including consent withdrawal, please see our Privacy Policy

Reviewer #1: No

Reviewer #2: **Yes: ** Renaud Condé

---

## [Editor Report · Acceptance letter]

PONE-D-25-16377R1

PLOS ONE

Dear Dr. Teillet,

I'm pleased to inform you that your manuscript has been deemed suitable for publication in PLOS ONE. Congratulations! Your manuscript is now being handed over to our production team.

Kind regards,

on behalf of

Dr. Humberto Lanz-Mendoza

Academic Editor

PLOS ONE